# Designing Carbon-Enriched Alumina Films Possessing Visible Light Absorption

**DOI:** 10.3390/ma15072700

**Published:** 2022-04-06

**Authors:** Arunas Jagminas, Vaclovas Klimas, Katsiaryna Chernyakova, Vitalija Jasulaitiene

**Affiliations:** State Research Institute Center for Physical Sciences and Technology, Sauletekio Ave. 3, 10257 Vilnius, Lithuania; vaclovas.klimas@ftmc.lt (V.K.); katsiaryna.chernyakova@ftmc.lt (K.C.); vitalija.jasulaitiene@ftmc.lt (V.J.)

**Keywords:** aluminum, anodizing, oxide films, heterostructure, carbon

## Abstract

Aluminum anodization in an aqueous solution of formic acid and sodium vanadate leads to the formation of alumina/carbon composite films. This process was optimized by varying the concentrations of formic acid and sodium vanadate, the pH, and the processing time in constant-voltage (60–100 V) or constant-current mode. As estimated, in this electrolyte, the anodizing conditions played a critical role in forming thick, nanoporous anodic films with surprisingly high carbon content up to 17 at.%. The morphology and composition of these films were examined by scanning electron microscopy, ellipsometry, EDS mapping, and thermogravimetry coupled with mass spectrometry. For the analysis of incorporated carbon species, X-ray photoelectron and Auger spectroscopies were applied, indicating the presence of carbon in both the *sp*^2^ and the *sp*^3^ states. For these films, the Tauc plots derived from the experimental diffuse reflectance spectra revealed an unprecedentedly low bandgap (*E*_g_) of 1.78 eV compared with the characteristic *E*_g_ values of alumina films formed in solutions of other carboxylic acids under conventional anodization conditions and visible-light absorption.

## 1. Introduction

The formation, structure, and properties of anodic aluminum oxide (AAO) films in aqueous solutions of sulfuric, oxalic, and *o*-phosphoric acids have been studied for decades. Aluminum anodization in sulfuric acid solution has a long history for the fabrication of protective coatings, whereas anodization processes in aqueous oxalic acid solution have been used since the 2000s, mainly for high-ordered porous alumina structure fabrication [1,2,3]. It is worth noting that, in these solutions, the content of entrapped oxalate anions inside the film framework at a typical bath voltage of 40–80 V is approximated at just 2 at.% [4]. Quite similar amounts of carbonaceous ions in different chemical states have also been determined in the alumina films formed by Al conventional anodization in tartaric, malonic, maleic, citric, and other carboxylic acid-containing solutions [5]. The content of entrapped electrolyte anions increased with the anodizing voltage value and decreased with the bath temperature [6,7]. To produce ordered anodic alumina with various morphologies, high-field anodization of Al has been proposed in oxalic [8,9,10], sulfuric [11,12], and phosphoric [13] acid solutions. For example, the growth of alumina films in a solution of 0.4 mol·L^−1^ tartaric acid at a high current density of 70 mA·cm^−2^ resulting in a steady-state anodizing voltage as high as 205 V was reported [14]. In this case, it was estimated that, in addition to water molecules and chemisorbed hydroxyl ions, several carbon-bearing species, namely, CO_2_, CO, COO^−^, CO_3_^2−^, and amorphous carbon, were entrapped with a total content of just 3.2 at.%. However, we demonstrated that, through the burning anodization of Al in the tartaric acid solution at critically high voltages, AAO films with an atypical nanotubed structure and entrapped carbon species content as high as 19.5 at.% can be designed [15]. A detailed study of these films by X-ray and Auger photoelectron spectroscopes indicated the entrapment of graphene oxide, resulting in a significant alumina bandgap value decrease to 2.37–1.63 eV depending on the bath concentration. In addition, these films were extremely hard and stable in acidic environments. However, due to especially high current density required for the creation of burning conditions, the formation of such films on large surfaces is problematic [15]. Furthermore, this process is hardly controllable due to an intense evolution of Joule heat from an electroconvection origin via oxidation/etching reactions at the AAOAl boundary [16] with chaos streaks [17].

In this study, to overcome this obstacle, we report a new method for the formation of a porous anodic aluminum (PAA) film heterostructure with carbon-containing species at a significantly lower anodization voltage and current density, and with a larger content of entrapped carbon species. This process allowed for the formation of thick, uniform, and colored alumina/carbon hybrid films on significantly larger surfaces compared with films formed previously in tartaric acid solution under burning conditions. We envisage that thick alumina films containing a high amount of entrapped carbonaceous species can be used for the absorption of microwaves and fabrication of invisible objects.

## 2. Materials and Methods

### 2.1. Materials

Deionized distilled water was used in the preparation of all solutions. Vanadic and formic acids were of analytical grade (Reachem Slovakia). Aluminum samples cut from high-purity (99.99 at.%, Goodfellow) foil with a thickness of 125 μm into specimens with a size of 17 × 17 mm^2^ were used. Prior to anodization, the specimens were etched in a solution of 1.5 mol·L^−1^ NaOH at 60 °C for 10 s, rinsed, desmutted in 1:1 H_2_O:HNO_3_ by sonication for 3 min, thoroughly rinsed again, and dried in an argon stream. The sample devoted to the ellipsometry test was electropolished in ethanol (EtOH) solution containing 1.5 mol·L^−1^ HClO_4_ (48%, Reachim, Moscow, Russia) and 120 mL·L^−1^ glycerol, thermostated at 3 ± 1 °C and 17 V for 90 s, and then rinsed with EtOH and water. Prior to anodization, this sample was etched in 0.24 mol·L^−1^ NaHCO_3_ at 80 °C for 1 min and then in 1 mol L^−1^ HNO_3_ solution for 30 s, rinsed in running and distilled water, and dried in N_2_ stream.

### 2.2. Anodization

For one-side anodization, a 250 mL Teflon cell was applied, featuring a circular window (area 0.785 cm^2^) at the bottom part for fixing the sample with a silicon ring. The aqueous solutions containing 0.2 to 1.0 mol·L^−1^ formic acid and 0.05 to 0.25 mol·L^−1^ NaVO_3_ were thermostated at 20 °C with a Lauda Alpha RA12 and gently mixed. A platinum sheet was used as the cathode, and Hg/Hg_2_SO_4_ in saturated K_2_SO_4_ solution was used as the reference electrode. Electrochemical investigations were conducted in a typical three-compartment electrochemical cell equipped with a Pt counter electrode, Hg/HgSO_4_/K_2_SO_4_(sat) reference electrode, and Al working electrode. The anodization was conducted at a constant potential up to 100 V. A ZAHNER-Elektrik GmbH&Co.KG (Kronach, Germany) Germany electrochemical workstation equipped with booster CVB 120 was used to control the potential and current generated. All potentials are presented on the SHE scale.

### 2.3. Characterization

The morphology and composition of the anodized samples were studied on both sides of the film and in their cross-section using an SEM FEI Helios Nanolab 650 equipped with a field-emission gun and EDX spectrometer (Eindhoven, The Netherlands). Cross-sections were obtained by brittle cracking of the film via bending of the sample over the blade of a knife. The thickness of thin films formed by anodic treatments in the formic and vanadic acids alone was determined by analyzing the corresponding ellipsometry plots, whereas the thickness of thicker films was determined by the SEM observations of their cross-sections. Prior to observations, the film surfaces were covered by sputtering of a thin Cr layer. Variations of *Ψ* and *Δ* parameters for nonporous alumina layers formed in the solutions of vanadate were determined in the region from 200 to 1700 nm wavelengths by spectroscopic ellipsometry. Bruggeman’s effective medium approximation was applied to calculate the parameters of these films using Complete EASE software. XPS measurements were performed to determine the chemical states of carbon, aluminum, vanadium, and oxygen and the surface composition of heterostructured films. XP spectra were acquired on an ESCALAB MKII (Thermo VG Scientific, East Grinstead, UK) spectrometer. All spectra were calibrated using the C_1*s*_ peak at 284.6 eV and analyzed using a nonlinear Shirley-type background. Recording and fitting of the spectra and calculation of the elemental composition were performed using Avantage software (5.918) provided by Thermo VG Scientific. 

The diffuse reflectance of the films was studied using a Shimadzu UV-VIS-NIR Spectrophotometer UV-3600 equipped with an integrating sphere in the range 200–1500 nm. Light absorbance and optical bandgaps were calculated using the Kubelka–Munk and Tauc functions, respectively.

To determine the content of carbon entrapped in the anodic film bulk, thermogravimetry (TG) analysis coupled with mass spectrometry (MS) of evolved CO_2_ and CO gases was applied. The simultaneous thermal analysis apparatus STA Pt 1600 (Linseis, Selb, Germany) was used for this together with the mass spectrometer MS Thermostar GDS 320 (Linseis/Pfeiffer, Asslar, Germany). For the investigation, 10–12 mg specimens were loaded in Al_2_O_3_ crucibles and then subsequently heated in an argon (Ar 6.0) and synthetic air (Ar 6.0–75; O_2_ 5.0–25 cm^3^·min^−1^) atmosphere up to 1000 °C at a heating rate of 10 °C·min^−1^. The data were collected and fitted by the software ‘Evaluation’ and ‘Quadera’ provided with the equipment.

## 3. Results and Discussion

### 3.1. Anodic Behavior of Aluminum in the Vanadate Solutions

Although vanadic acid (HVO_3_) and vanadate solutions are well known as corrosion inhibitors of Al and its alloys [18,19], the anodic behavior of Al in these solutions remains unclear. Despite several reports, detailed studies are scarce. We suspect that, due to a low solubility of vanadic acid, neutral and slightly acidic aqueous solutions might be a prerequisite in constructing their electrolytes for Al anodization. It is worth noting that, in acidic solutions, vanadic acid transforms to a *meta* state, which further tends to polymerize into di-, tetra-, and decavanadic acids depending on the pH value [20]. The current transience, *i*_a_(*t*), (Figure 1a) as well as anodic potential–time, *E*_a_(*t*), plots (Figure 1b) recorded for Al anodized in pure 0.2 mol·L^−1^ sodium vanadate solution verified the formation of a barrier-layer film [21]. The thickness of this layer (*δ*_b_) determined by ellipsometry was about 116–121 nm. In the case of *E*_a_ = 80 V, the anodic oxidation rate was 1.45–1.5 nm·V^−1^, which is close to the 1.4 nm·V^−1^ rate characteristic of barrier-layer alumina film formation. We also determined that, for a given anodization time, *E*_a_(*t*) plots were linear (Figure 1c) with the slope varying to some extent with the pH of the solution applied. For example, if the pH increased from 7.1 to 8.4, the slope of the *E*_a_(*i*) plots decreased from 0.257 + 0.0005*i*_a_^−1^ to 0.221 + 0.003 *i*_a_^−1^. We determined that a linear growth of alumina film at a constant current density proceeded up to 178–185 V. With further processing, the breakdown of the film began to cause a drop in *E*_a_ to 163–167 V along with oscillations.

### 3.2. Anodic Behavior of Aluminum in HCOOH Solutions

It has been reported that anodic films thicker than 100 nm cannot be formed on the Al surface in monobasic acid solutions, such as HCl and HNO_3_ [22]. However, opposite information can be found in the available literature about the anodic behavior of Al in aqueous solutions of formic acid. According to Tajima [22], anodic treatment of Al in HCOOH solution resulted mainly in pitting-like surface etching followed by the formation of a thin oxide film. It has also been reported that, in diluted and low-temperature solutions, local corrosion and pitting of the aluminum surface prevail, whereas, at relatively high temperatures and concentrations, porous oxide layers with an irregular structure within a wide range of anodic voltages can be formed [23,24]. Moreover, Pashchanka and Schneider [25] reported that, in room-temperature solutions, porous alumina with a clear tendency to hexagonal order was formed at 18–30 V. In this study, we also investigated the anodic behavior of aluminum in aqueous formate solutions. From these investigations, several conclusions were drawn: (i) a sudden increase in potential toward the maximum value at the onset of galvanostatic anodization is characteristic of pH within the 1.8–8.4 range; (ii) with further processing at a constant *i*_a_, *E*_a_ decreased to a similar value independent of the electrolyte pH. For example, at *i*_a_, = 5 mA cm^−2^ and pH = 1.8, *E*_a_ stabilized at 24–20 V; (iii) similar behavior was estimated in the slightly alkaline solutions, revealing a rapid *E*_a_ stabilization at some lower voltages, e.g., 20–21 V, whereas, in the slightly acidic solutions, the *E*_a_ vs. *t* plots varied significantly, not stabilizing for several minutes; (iv) anodization at a steady-state potential for hours resulted in the formation of porous anodic films with sub-micrometer thickness.

### 3.3. Anodic Behavior of Aluminum in NaVO_3_–HCOOH–HCOONa Solution

To optimize Al anodization in this composite solution, we investigated a number of process variations. It was established that the addition of just 0.01 mol·L^−1^ sodium vanadate to sodium formate solution resulted in an increase in steady-state anodizing potential from 24–25 V to 40–42 V (Figure 2a). A further increase in NaVO_3_ concentration to 0.2 mol·L^−1^ was not so effective (Figure 2a, plots 3 and 4) compared to 0.01 mol·L^−1^ addition. Figure 2b displays the *E*_a_(*t*) plots recorded in a solution of 0.25 mol·L^−1^ NaVO_3_ without (1) and with various NaCO_2_H contents at a constant current density of 0.5 mA·cm^−2^. According to these plots, the concentration of formate is crucial for the anodic behavior of Al under galvanostatic conditions; with an increase in the formate concentration, the growth of the film after barrier-layer formation (star in Figure 2b) proceeded with a decreasing rate of potential. In the case of potentiostatic anodization (Figure 2c), an increase in vanadate concentration to 0.05 mol·L^−1^ resulted in a sudden decrease in current density during first 100 s to steady-state growth without any current fluctuations characteristic of pure formate solutions. Lastly, it was determined that an increase in the content of sodium vanadate in the mixed electrolyte of formic acid/sodium formate always resulted in an increase in alumina formation *E*_a_ and a decrease in the *i*_a_, while a decrease in the pH of electrolyte led to film formation at a lower potential.

### 3.4. Films

It was established that a key parameter influencing the formation of PAA film on the Al surface in the mixed electrolyte was the pH. Slightly colored films could be formed within the pH range of 2.5–5.6 starting from *E*_a_ = 23 V. However, the as-formed films, even after 3–4 h of Al anodization at this voltage, remained nonuniformly colored in light tones. Films colored in khaki and black spots started to form at 35–38 V. Uniform and deep-colored finishing of the Al surface could be obtained within a wide range of anodizing voltages, e.g., at 40–180 V using 0.2–0.25 NaVO_3_ and 0.5–1.0 mol·L^−1^ HCOOH solution with pH from 2.5 to 3.0. For example, at 20 °C and 80 V, deep-colored films with a highly uniform thickness of ~40 μm were formed after 1 h of anodization. These films dissolved in NaOH forming a brownish solution. We suspect that the color of the solution cannot be ascribed to the presence of V^5+^ (colorless) or V^4+^ (blue) compounds [26].

The structure of PAA films formed after 60 min in the optimized composite solution (0.2 mol·L^−1^ NaVO_3_ + 0.8 mol·L^−1^ HCOOH; pH 2.6) was further studied for three values of *E*_a_, namely, 60, 80, and 100 V. Both sides of the films and their cross-sections were inspected. Figure 3 displays the SEM images of films obtained by the prolonged anodization of Al at 60 and 80 V. From these images, the dense packaging of honeycomb-shaped cells can be verified. The size and uniformity of cells at the metal–film interface varied significantly. The uniformity and ordering of cells seemed to be poor for all tested *E*_a_ modes. In addition, in the case of thinner films formed after several minutes, groups of obviously larger cells randomly distributed and uneven in size can be seen (Figure 3a). More detailed inspection of the areas with larger cells on both film sides revealed the formation of 2–3-fold wider pores in these areas and twofold greater film morphology after prolonged Al anodization (Figure 3b). The formation of films consisting of uneven and larger cells with wide pores at the Al–film interface and a continuous thick upper layer with threefold thinner pores can be observed after specimen bending and layer cleavage (Figure 3b). We found, however, that if anodization was initiated during the first 2–5 min at a lower voltage of 30–40 V, a further stepwise increase in *U*_a_ to 60–100 V caused the formation of single-layered alumina films with quite uniform pores across all film thicknesses, which were not cleaved upon specimen bending (Figure 3d,e). It is also worth noting that the thickness of PAA film in these solutions surprisingly depended on the increase in anodization potential value with the decrease in *E*_a_ from 100 to 60 V (Figure 3c,f). According to Pashchanka [25], the electroconvection-based mechanism underlying the growth of such films, known as ion exchange, can be attributed to nonstoichiometric amphoteric amorphous alumina formation in an acidic environment, resulting in nonuniform distribution of impurity-related elements inside the pore walls [27,28].

### 3.5. The Composition of Films 

Chemical analysis using XPS indicated the presence of Al, O, V, and C elements. The quantification of XPS peaks of Al samples anodized in the optimized solution (0.2 mol·L^−1^ NaVO_3_ + 0.8 mol·L^−1^ HCOOH; pH 2.67) at 60, 80, and 100 V for 60 min is presented in Table 1. According to these data, quite similar contents of elements were determined regardless of the *E*_a_ value. The content of encased oxygen was 11–12% higher than that required for Al_2_O_3_ and VO_3_ stoichiometry, implying the inclusion of water molecules, as well as OH^−^ and HCOO^−^ anions. The content of entrapped vanadium was low, 2.48 ± 0.22 at.%, insignificantly increasing with *E*_a_ value. Furthermore, a surprisingly high amount of entrapped carbon, ca. 17 at.%, regardless of *E*_a_, was detected for all specimens. The observed phenomenon contradicted the well-known rule for incorporating electrolyte anions inside the outer layer of porous alumina films postulating that the content of incorporated electrolyte anions should increase with anodizing voltage and decrease with increasing electrolyte temperature [29]. Of note, such a high content of carbonaceous species was observed on the surface of alumina films formed in tartaric acid solution at the critical voltages and extremely high current densities causing burning [15]. The three peaks (Figure 4) with binding energies (BEs) of 515.39–515.66, 516.54–516.84, and 517.59–517.71 eV were attributable to V^3+^, V^4+^, and V^5+^ states, respectively [30]. 

Quantification of their peak areas (see Table 1) revealed the dominant presence of vanadium in the +4 oxidation state, whereas the content of entrapped V^5+^ species decreased with Ea (Table 2).

Deconvoluted O_1*s*_ spectra were found to be described by three counterparts with BE peaks within 530.30–530.35, 531.56–531.61, and 532.73–532.90 eV ranges, corresponding well to the BEs of oxygen in oxides, hydroxides, and carbohydrates, respectively [31]. For all specimens, the XPS spectra of C_1*s*_ were deconvoluted into four counterparts (Figure 5) attributable to C–C, C–OH, C–O, and carboxylic O=C–O bonds with BEs of 284.49–284.60 eV, 285.68–286.03 eV, 286.72–287.13 eV, and 289.01–289.12 eV, respectively [32,33]. From the quantification of peak areas, most carbonaceous species could be ascribed to C–C bonds, while the smallest fraction of entrapped carbon was characteristic of C–O bonded species. Quantitative analysis data are presented in Table 3. 

Furthermore, the *D* parameter calculated from the differentiated *dC/dE* versus kinetic energy plots for various *E*_a_ samples on the surface was equal to 12.8–14.0 eV, implying a diamond-like carbon nature [33]. This was further confirmed by the carbon *KVV* spectra acquired with a low-energy electron beam (3 keV) showing *D* = 20.4 eV. According to [32,33,34], the shift in *D* parameter between X-ray and electron beam-induced Auger spectra represents the fingerprint of graphene. It is worth noting that, from the C_1*s*_ deconvoluted areas, the content of carbon entrapped inside the anodic film in the graphene state was the highest among all carbonaceous species regardless of *E*_a_ value. 

To determine the average content of carbon entrapped inside the anodic film formed in a solution containing 0.8 mol·L^−1^ HCOOH and 0.2 mol·L^−1^ NaVO_3_ (pH 2.7) at 80 V for 1 h, the probe of this film with a thickness of ~42 μm was analyzed further by thermogravimetry analysis coupled with mass spectrometry up to 1000 °C. To determine the content of remaining carbon after the TG test in argon, we burned this probe in a synthetic air atmosphere by heating to 1000 °C. The results obtained are shown in Figure 6. According to this plot, the average amount of carbon in the film was approximated at 7.7 at.%, and it was removed from the film in the form of CO_2_ and CO gases at about 900 °C. Furthermore, 94.8% of all carbon evolved during heating in an argon atmosphere. A relatively uniform distribution of carbon species in the film bulk was also elucidated by EDS mapping of carbon element in the film cross-section, evidencing the formation of a hybrid alumina–carbon material (Figure 7c,e).

The large content of entrapped carbonaceous species in the anodic alumina films grown at 60–100 V was established for the first time. It appears that these species are inserted into the alumina cell bulk during film growth in a similar way to that seen for structural ions in other electrolytes, coloring the film from grayish to black with an increase in anodization time and film thickness. The main carbon source might be COO^−^ ions, supplied through the barrier layer during anodization to the Al–oxide interface together with OH^−^/O^2–^ ions. From the large content of carbon entrapped in the thin alumina films (5.8 at.%) formed in the formate solutions, it follows that formate ions were transported easily through the alumina barrier layer and reacted with Al^3+^ species at the Al–oxide interface.

These formate ions are lighter compared to the twofold larger oxalate ones in the solution of oxalic acid. Contaminations extracted from the XPS C_1*s*_ peak deconvolution showed that the main source of entrapped carbon is the graphite C–C bond. The inserted HCOO^−^ ions accounted for just one-fourth of the total carbon. However, the exact decomposition mechanism of formate ions and the insertion pathway of carbonaceous species require further investigation. These questions will be addressed in future work.

### 3.6. Optical Properties

The anodic oxides of Al obtained in sulfuric and phosphoric acid solutions are transparent and colorless, while those grown in oxalic, citric, and tartaric acid solutions are slightly yellow, greenish gray, and gray, respectively [35]. The optical bandgap energy (*E*_g_) of conventional Al oxides allows direct transitions from 5.40 eV to 5.75 eV [36]. The *E*_g_ of anodic Al oxides ranges from 1.6 eV [15] to 4.3 eV [37]. Furthermore, the *E*_g_ value of oxalic acid anodic films slightly increases with anodizing voltage but remains constant with an increase in film thickness and porosity [38]. As reported in [37,39], photoluminescence investigations indicated the presence of F and F^+^ centers in oxalic and sulfuric alumina films, which could be attributed to oxygen vacancy-related defect centers. The diffuse reflection spectra and the corresponding Tauc plots of alumina films formed in the optimized vanadate–formate solutions were found to be dependent on the anodization time (Figure 8). Moreover, the films formed at *E*_a_ values from 60 to 100 V for the same time, e.g., 1 h, possess a similar *E*_g_ value of about 1.78 eV, regardless of the *E*_a_ value, which is in the far-visible region (Figure 8c, plots 1 to 3). Significantly, this band value is lower than that reported for porous alumina films formed in sulfuric, oxalic, or carboxylic acid solutions under conventional conditions [37]. However, in the case of thinner films, obtained, for example, during 2 and 5 min anodization at a stable potential of 80 V, their light absorption edge was approximated at 3.19 and 2.70 eV (Figure 8c, 4 and 5 plots), respectively, implying the potential to control the *E*_g_ value and absorption properties of these films as a function of anodization time.

Our electrolyte contained an organic acid (HCOOH) and vanadate. Therefore, it is reasonable to suggest that entrapped vanadium species could influence the optical properties of these hybrid films. According to previous reports [40,41], the bandgap energies of vanadium oxides O_2*p*_–V_3*d*_ were equal to 2.0–2.6 eV. Therefore, the calculated *E*_g_ values in the range 3.05–3.20 eV would not affect the absorption of films formed in our electrolyte after 2 min with shorter processing, whereas the entrapment of vanadium oxides in thicker films with *E*_g_ = 2.98–3.1 eV could have been a factor. Another factor capable of distorting the true *E*_g_ value of our films according to the diffuse reflectance spectral analysis via the Tauc approach is the Al layer remaining under the film capable of absorbing 1.5 eV energy [42]. To avoid this, we anodized 25 μm thick Al foil and analyzed the obtained transmission spectra before and after film calcination at 950 °C for 2 h (Figure 9). It is worth noting that the calculated *E*_g_ value for this and the Al substrate film grown at 80 V for 1 h showed quite a similar *E*_g_ value of 1.78 eV for the direct forbidden transition. However, the calcination of aluminum oxide/carbon hybrid material in air increased *E*_g_ by 1.2–1.35 eV (see Figure 9).

## 4. Conclusions

The anodic behavior of aluminum in aqueous solutions of formic and/or vanadic acid electrolytes was studied. A complete change in anodic behavior of Al from the intense and uneven surface oxidation/etching in HCOOH and barrier-type film formation in HVO_3_ to the formation of a thick and nanoporous film in the mixed solution was observed. The optimal composition of the solution and anodization conditions were established on the basis of the *i*_a_(*t*) transience and the corresponding film morphology. These investigations revealed the incorporation of a low content of vanadium species in the V^5+^, V^4+^, and V^3+^ states and large amounts of carbonaceous species into the oxide framework, ca. up to 17 at.%, regardless of anodizing potential value within the 60 to 100 V range. This, in turn, led to the formation of a black anodic film with an atypically low bandgap value of 1.78 eV and the capability to absorb visible light. The XPS and Auger electron spectroscopy investigations indicated the inclusion of carbonaceous species in two different C–C bond hybridization states. The *D* parameter calculated from the differentiated *d**C*/*d**E* versus kinetic energy plots for various *E*_a_ samples on their surface side was determined to be approximately 12.8–14.0 eV, implying a diamond-like carbon nature. In contrast, the carbon Auger *KVV* spectra acquired with a low-energy electron beam (3 keV) revealed *D* = 20.4 eV, indicative of the presence of graphene. However, additional experiments are required to precisely establish the ratio of entrapped *sp*^2^/*sp*^3^ carbon species with the BE range 284.49–284.6 eV, characteristic of C_1*s*_ diamond and graphitic states using synchrotron radiation. The cost-effective and straightforward method reported herein for alumina/carbon composite film formation with a surprisingly low bandgap opens new horizons for their application as microwave absorbers and materials for invisible devices.

## Figures and Tables

**Figure 1 materials-15-02700-f001:**
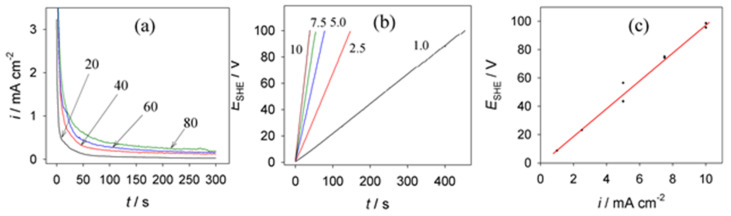
Variations in current transience (**a**) and Al anodizing voltage (**b**) with processing time at the indicated voltage (**a**) and current density (**b**) values in 0.2 mol·L^−1^ NaVO_3_, pH 7.1. (**c**) Typical voltage–current plot for 0.2 mol·L^−1^ NaVO_3_ slightly alkaline (pH 8.4) solution.

**Figure 2 materials-15-02700-f002:**
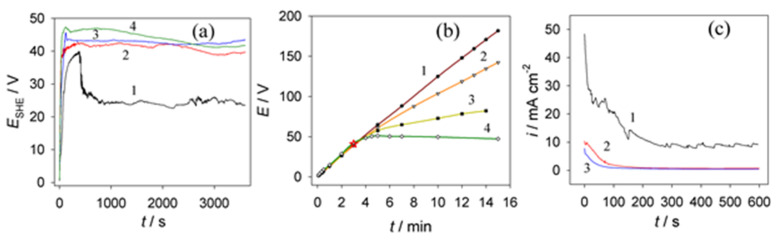
(**a**) *E*_a_ vs. *t* plots for 1.0 mol·L^−1^ formic acid solution containing 0 (1), 0.01 (2), 0.05 (3), and 0.2 mol·L^−1^ NaVO_3_ at a pH 2.6, 20 °C, and *i*_a_ = 5 mA·cm^−1^. (**b**) Same as (**a**) at a current density of 0.5 mA·cm^−2^ in a solution containing 0.25 mol·L^−1^ NaVO_3_ without (1) and with 0.1 (2) or 1.0 mol·L^−1^ (3,4) NaCO_2_H at 20 °C; pH 8.4 (1–3) and 2.7 (4). (**c**) *i*_a_(*t*) plots in 1.0 mol·L^−1^ CO_2_H_2_ + NaCO_2_H solution showing the influence of adding (1) 0, (2) 0.05, and (3) 0.2 mol·L^−1^ NaVO_3_; pH 2.6, 20 °C, *E*_a_ = 37 V.

**Figure 3 materials-15-02700-f003:**
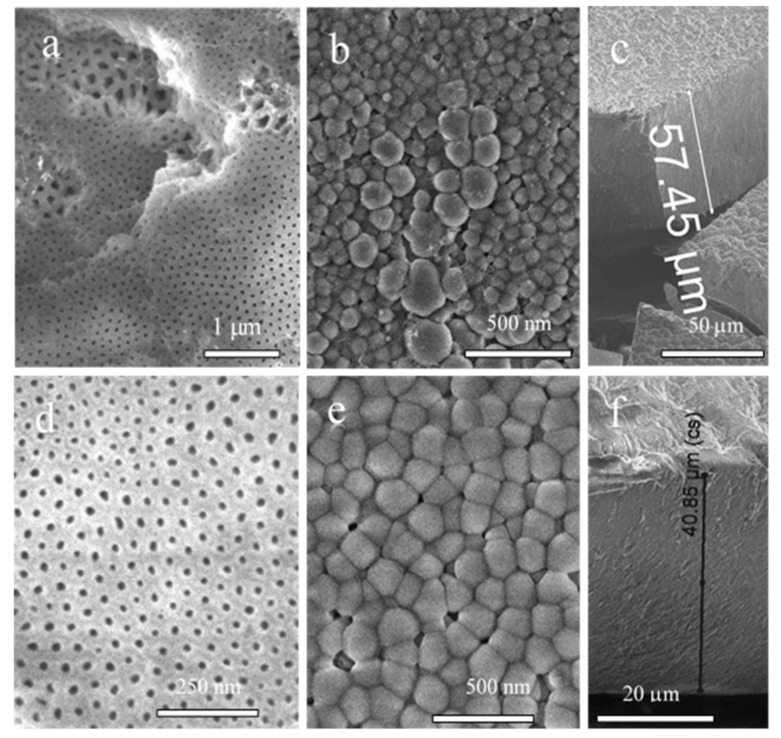
Top-side (**a**,**d**), back-side (**b**,**e**), and cross-sectional (**c**,**f**) SEM images of anodic film formed from Al anodization in a solution containing 0.8 mol·L^−1^ HCOOH and 0.2 mol·L^−1^ NaVO_3_ (pH 2.7; 20 °C) at 60 V (**a**–**c**) and 80 V (**d**–**f**) for 1 h.

**Figure 4 materials-15-02700-f004:**
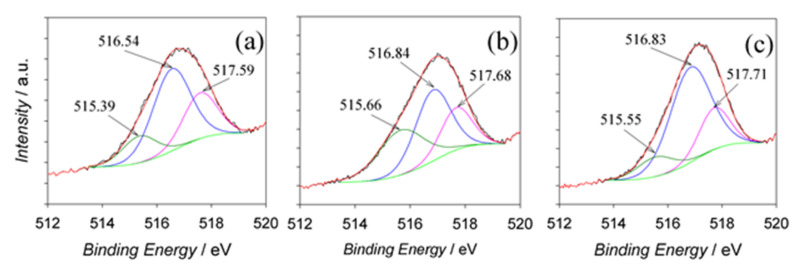
Deconvoluted V2*_p3/2_* XPS spectra of vanadium species entrapped inside the films fabricated by Al anodization in a solution containing 0.2 mol·L^−1^ NaVO3 and 0.8 mol·L^−1^ HCOOH (pH 2.5) at *E*_a_ values of (**a**) 60, (**b**) 80, and (**c**) 100 V for 60 min.

**Figure 5 materials-15-02700-f005:**
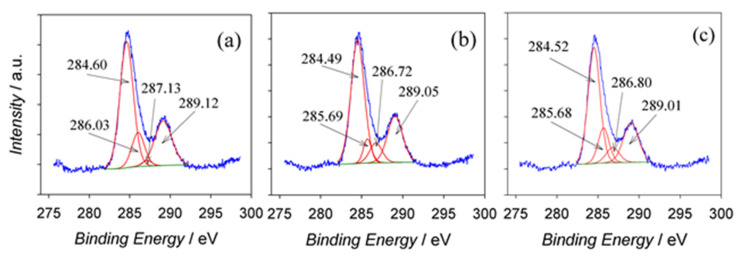
Deconvoluted C_1*s*_ XPS spectra of carbonaceous species entrapped inside the films fabricated by Al anodization at *E*_a_ values of 60 (**a**), 80 (**b**), and 100 (**c**) V for 60 min.

**Figure 6 materials-15-02700-f006:**
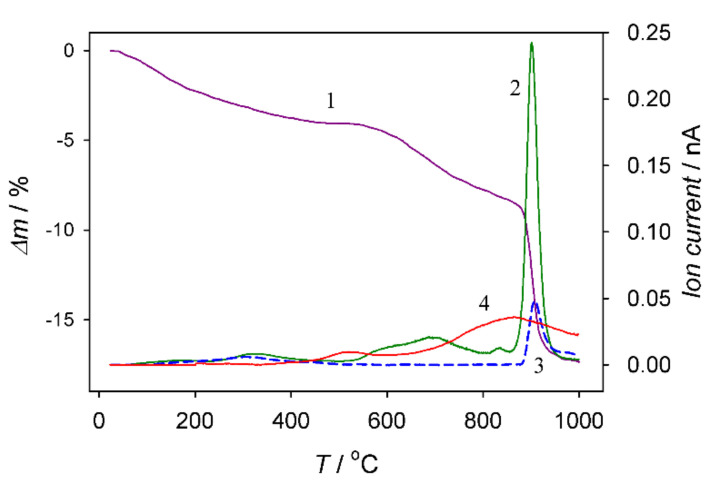
Variation of TG curve (1) and ionic currents of CO_2_ (2) and CO species (3) during calcination of ~42 μm thick aluminum anodic film in an argon atmosphere up to 1000 °C and then in synthetic air up to 1000 °C for the burning of remaining carbon to CO_2_ (plot 4, × 10). The anodic film was formed at 80 V for 1 h as in Figure 3.

**Figure 7 materials-15-02700-f007:**
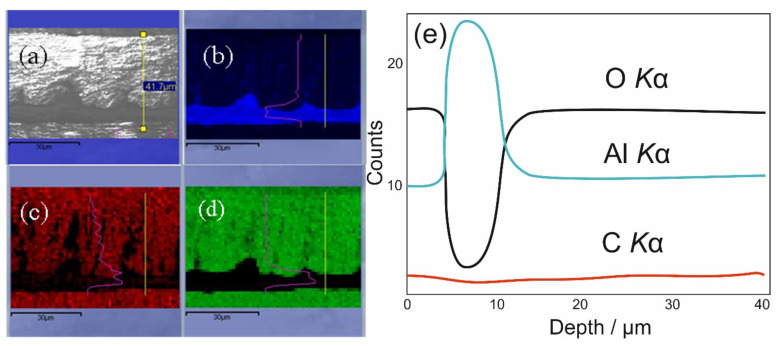
Cross-sectional image of AAO/carbon composite film (**a**) and EDS mapping of aluminum (**b**), carbon (**c**), and oxygen (**d**) elements. (**e**) Cross-sectional distribution curves of oxygen, aluminum, and carbon elements in the film formed as in Figure 3 at 80 V.

**Figure 8 materials-15-02700-f008:**
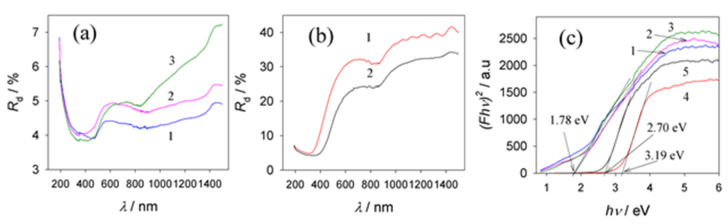
Diffuse reflectance spectra (**a**) and corresponding Tauc plots for possible direct transitions (**c**) in the composite films fabricated by Al anodization in a solution containing 0.2 mol·L^−1^ NaVO_3_ and 0.8 mol·L^−1^ HCOOH (pH 2.67) at bath voltages of 60 (1), 80 V (2), and 100 V (3) for 1 h. (**b**) As in (**a**) for Al anodization at 80 V for 2 (1) and 5 (2) min. (**c**) Tauc plots for possible direct transitions in the composite films fabricated by Al anodization in the same solution at bath voltages of 60 (1), 80 (2), and 100 V (3) for 1 h, and at 80 V for 2 (4) and 5 min (5).

**Figure 9 materials-15-02700-f009:**
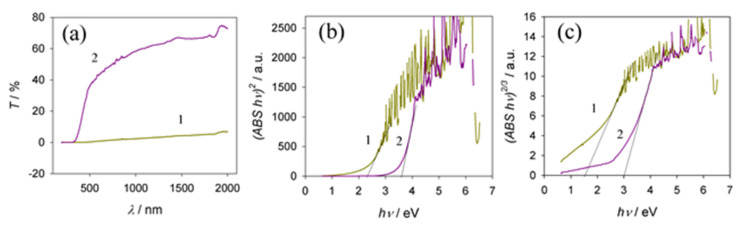
The transmission spectra (**a**) and calculated Tauc dependencies for direct (**b**) and direct forbidden transitions (**c**) in aluminum oxide samples obtained by Al anodization in 0.2 mol·L^−1^ NaVO_3_ + 0.8 mol·L^−1^ HCOOH electrolyte (pH = 2.6) at 80 V before (1) and after calcination at 950 °C for 2 h (2).

**Table 1 materials-15-02700-t001:** XPS data for anodic films fabricated by Al anodization in a solution containing 0.2 mol·L^−1^ NaVO_3_ and 0.8 mol·L^−1^ HCOOH (pH 2.5) at the indicated bath voltages, *E*_a_, for 60 min.

*E*_a_, V	Name	Peak BE	FWHM eV	Area (P) CPS.eV	Atomic %	Q	SF
60	V_2*p*_	516.74	2.52	25,150	2.26	1	6.330
	O_1*s*_	531.66	3.17	267,785	53.51	1	2.850
	C_1*s*_	284.73	2.28	30,885	17.08	1	1.000
-	Al_2*p*_	74.24	2.04	28,750	27.16	1	0.574
80	V_2*p*_	516.89	2.57	29,080	2.45	1	6.330
-	O_1*s*_	531.61	3.19	281,965	52.80	1	2.850
-	C_1*s*_	284.59	2.19	33,005	17.11	1	1.000
-	Al_2*p*_	74.19	2.09	31,225	27.64	1	0.574
100	V_2*p*_	517.04	2.34	29,354	2.71	1	6.330
-	O_1*s*_	531.70	3.20	261,951	53.75	1	2.850
-	C_1*s*_	284.67	2.19	30,490	17.31	1	1.000
-	Al_2*p*_	74.27	2.03	27,040	26.23	1	0.574

**Table 2 materials-15-02700-t002:** The relative proportions of vanadium species according to V_2*p*3/2_ spectrograms of the films fabricated as in Table 1.

**Anodizing Voltage, V**	**V^3+^**	**V^4+^**	**V^5+^**
60	0.166	0.555	0.278
80	0.353	0.423	0.224
100	0.121	0.663	0.215

**Table 3 materials-15-02700-t003:** The relative proportions of carbon species according to C_1*s*_ spectrograms of the films fabricated as in Table 1.

**Anodizing Voltage, V**	**284.49–284.60 eV** **(C–C, C–H)**	**285.68–286.03 eV** **(C–OH)**	**286.72–287.13 eV** **(C–O)**	**289.01–289.12 eV** **(C–OOH)**
60	0.572	0.146	0.019	0.263
80	0.589	0.070	0.090	0.253
100	0.539	0.149	0.077	0.235

## Data Availability

Not applicable.

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
