# Peer review of "Designing Carbon-Enriched Alumina Films Possessing Visible Light Absorption"

_materials, 2022, doi:10.3390/ma15072700_

Round 1

Reviewer 1 Report

The paper discuss the development and properties of carbon enriched alumina films, the research findings are interesting and promising for future research in the field. The overall evaluation of the paper is good, but I recommend several changes beforeto consider it suitable for publication in the journal.

  • Title and abstract do not match precisely. For instance, in the title, the light absorption is mentioned, whilst it is not properly highlighted in the abstract. 
  • The introduction must be enriched discussing some applications of this type of materials, with proper references.
  • After the Introduction, Materials and Methods section must be placed in order to introduce the reader to the successive description of the results obtained.
  • Figure 1, panel a (left), shows some curves which are actually not distinguishable, please provide a suitable enlargement inside the graph space and consider to enlarge the figures and to modify their location (vertical rather than horizontal) if there is not enough space.
  • Section 2.2, row 85, the authors say "It is commonly accepted that anodic films thicker than 100 nm on the Al surface cannot be formed in the monocarboxylic acid solutions." Please, provide enough literature to support the stated assertion.
  • In section 2.3 consider to rewrite in a clearer way the part from row 106 to row 126, because it is rather hard to understand. 
  • In Figure 5 please remove C1s in the three panels, because it is unnecessary. On the other hand, authors must provide a legend, at least in one panel.
  • In Figure 6 please, use also different line types (dashed, dotted, etc...) to better distinguish the curves and use a legend to help the reader and to make more direct the related caption.
  • Figure 8 must be redrawn. In my opinion the curves obtained with different settings should belong at least to different panels. For instance 2 and 2' curves should be drawn separately. The labels themselves 2 and 2' are badly related to the parameters varied. Please, consider to separate that curves and to use legend in which must be clear the parameter chenged and, of course its, value.
  • In the conclusions, consider to recall the improved C concentration obtained, since it is one of the most interesting features described in the paper.
  • Again, section 4 (materials and methods) should be located after the introduction and before the presentation and discussion of the results.

Author Response

Response to Reviewer #1:  

The paper discuss the development and properties of carbon enriched alumina films, the research findings are interesting and promising for future research in the field. The overall evaluation of the paper is good, but I recommend several changes before to consider it suitable for publication in the journal.

  • Title and abstract do not match precisely. For instance, in the title, the light absorption is mentioned, whilst it is not properly highlighted in the abstract. 

Response: The light absorption of composite film was highlighted in the Abstract and Conclusions parts.

  • The introduction must be enriched discussing some applications of this type of materials, with proper references.

Response: Possible applications of Al porous anodic films designed in this study introduced in the ‘Introduction’ and ‘Conclusion’ parts.

  • After the Introduction, Materials and Methods section must be placed in order to introduce the reader to the successive description of the results obtained.

Response: The Materials and Methods section was placed after ‘Introduction’ part.

  • Figure 1, panel a (left), shows some curves which are actually not distinguishable, please provide a suitable enlargement inside the graph space and consider to enlarge the figures and to modify their location (vertical rather than horizontal) if there is not enough space.

Response: Fig. 1a part was modified by enlargement inside the graph space, as required.

  • Section 2.2, row 85, the authors say "It is commonly accepted that anodic films thicker than 100 nm on the Al surface cannot be formed in the monocarboxylic acid solutions." Please, provide enough literature to support the stated assertion.

Response: We specified this information:  It has been reported that anodic films thicker than 100 nm cannot be formed on the Al surface in the monobasic acid solutions, such as HCl and HNO3 [22].

  • In section 2.3 consider to rewrite in a clearer way the part from row 106 to row 126, because it is rather hard to understand. 

Response: The indicated part was rewrite clearer. The legend of Figure 2 – specified.

  • In Figure 5 please remove C1s in the three panels, because it is unnecessary. On the other hand, authors must provide a legend, at least in one panel.

Response: Indications ‘C1s’ were removed, as required.

  • In Figure 6 please, use also different line types (dashed, dotted, etc...) to better distinguish the curves and use a legend to help the reader and to make more direct the related caption.

Response: TG curves presented in Figure 6 differ in color. For clarity the curve no 3 was dashed.

  • Figure 8 must be redrawn. In my opinion the curves obtained with different settings should belong at least to different panels. For instance, 2 and 2' curves should be drawn separately. The labels themselves 2 and 2' are badly related to the parameters varied. Please, consider to separate that curves and to use legend in which must be clear the parameter changed and, of course its, value.

Response: Figure 8 was redrawn. The results are presented in three panels instead of two.

  • In the conclusions, consider to recall the improved C concentration obtained, since it is one of the most interesting features described in the paper.

Response: We add this information in the Conclusions part.

Reviewer 2 Report

The bulk alumina always possesses a very high bandgap and shows no applications in visible light. A. Jagminas et al have successfully prepared alumina films with visible light absorption. Present work is interesting and very well supported by characterization methods. I recommend this work for publication.

Major comments:

  • How do authors calculate at% from XPS? It would be helpful for readers to see any mathematical expression associated with the calculation.
  • Any visible light associated applications will enhance the quality of the further.

Minor comments:

  • Please take care of subscripts and superscripts throughout the manuscript (Ex: line 70, 76, 77, etc)
  • All references must be in square brackets. check line 93.

Author Response

Response to Reviewer #2:

The bulk alumina always possesses a very high bandgap and shows no applications in visible light. A. Jagminas et al have successfully prepared alumina films with visible light absorption. Present work is interesting and very well supported by characterization methods. I recommend this work for publication.

Major comments:

  • How do authors calculate at% from XPS? It would be helpful for readers to see any mathematical expression associated with the calculation.

Response: For clarity we placed the following info in the Experimental part: The spectra calibration, processing and fitting routines were done using Avantage software (5.918) provided by Thermo VG Scientific. Core level peaks of Al2p, V2p, C1s and O1s were analyzed using a nonlinear Shirley-type background and the calculation of the elemental composition was performed on the basis of Scofield’s relative sensitivity factors. This data system has an integrated library of peak positions and relative intensities. Identified peaks are automatically added to a peak table which can then be used for quantification. For quantification purposes, the data system has integral libraries of relative sensitivity factors (both Scofield and Wagner) and all data files contain information about the acquisition conditions, including the spectrometer conditions. The quantification routine automatically applies the correct instrument transmission function to the data.

  • Any visible light associated applications will enhance the quality of the further.

Response: Visible light associated applications of alumina-carbon films not studied yet. We address these investigations in future.

Minor comments:

  • Please take care of subscripts and superscripts throughout the manuscript (Ex: line 70, 76, 77, etc)

Response: The indicated mistakes corrected. Thank you for this remark.

  • All references must be in square brackets. check line 93.

Response: This mistake corrected to [23,24]

Response to Reviewer #2:

The bulk alumina always possesses a very high bandgap and shows no applications in visible light. A. Jagminas et al have successfully prepared alumina films with visible light absorption. Present work is interesting and very well supported by characterization methods. I recommend this work for publication.

Major comments:

  • How do authors calculate at% from XPS? It would be helpful for readers to see any mathematical expression associated with the calculation.

Response: For clarity we placed the following info in the Experimental part: The spectra calibration, processing and fitting routines were done using Avantage software (5.918) provided by Thermo VG Scientific. Core level peaks of Al2p, V2p, C1s and O1s were analyzed using a nonlinear Shirley-type background and the calculation of the elemental composition was performed on the basis of Scofield’s relative sensitivity factors. This data system has an integrated library of peak positions and relative intensities. Identified peaks are automatically added to a peak table which can then be used for quantification. For quantification purposes, the data system has integral libraries of relative sensitivity factors (both Scofield and Wagner) and all data files contain information about the acquisition conditions, including the spectrometer conditions. The quantification routine automatically applies the correct instrument transmission function to the data.

  • Any visible light associated applications will enhance the quality of the further.

Response: Visible light associated applications of alumina-carbon films not studied yet. We address these investigations in future.

Minor comments:

  • Please take care of subscripts and superscripts throughout the manuscript (Ex: line 70, 76, 77, etc)

Response: The indicated mistakes corrected. Thank you for this remark.

  • All references must be in square brackets. check line 93.

Response: This mistake corrected to [23,24]

Response to Reviewer #2:

The bulk alumina always possesses a very high bandgap and shows no applications in visible light. A. Jagminas et al have successfully prepared alumina films with visible light absorption. Present work is interesting and very well supported by characterization methods. I recommend this work for publication.

Major comments:

  • How do authors calculate at% from XPS? It would be helpful for readers to see any mathematical expression associated with the calculation.

Response: For clarity we placed the following info in the Experimental part: The spectra calibration, processing and fitting routines were done using Avantage software (5.918) provided by Thermo VG Scientific. Core level peaks of Al2p, V2p, C1s and O1s were analyzed using a nonlinear Shirley-type background and the calculation of the elemental composition was performed on the basis of Scofield’s relative sensitivity factors. This data system has an integrated library of peak positions and relative intensities. Identified peaks are automatically added to a peak table which can then be used for quantification. For quantification purposes, the data system has integral libraries of relative sensitivity factors (both Scofield and Wagner) and all data files contain information about the acquisition conditions, including the spectrometer conditions. The quantification routine automatically applies the correct instrument transmission function to the data.

  • Any visible light associated applications will enhance the quality of the further.

Response: Visible light associated applications of alumina-carbon films not studied yet. We address these investigations in future.

Minor comments:

  • Please take care of subscripts and superscripts throughout the manuscript (Ex: line 70, 76, 77, etc)

Response: The indicated mistakes corrected. Thank you for this remark.

  • All references must be in square brackets. check line 93.

Response: This mistake corrected to [23,24]

Response to Reviewer #2:

The bulk alumina always possesses a very high bandgap and shows no applications in visible light. A. Jagminas et al have successfully prepared alumina films with visible light absorption. Present work is interesting and very well supported by characterization methods. I recommend this work for publication.

Major comments:

  • How do authors calculate at% from XPS? It would be helpful for readers to see any mathematical expression associated with the calculation.

Response: For clarity we placed the following info in the Experimental part: The spectra calibration, processing and fitting routines were done using Avantage software (5.918) provided by Thermo VG Scientific. Core level peaks of Al2p, V2p, C1s and O1s were analyzed using a nonlinear Shirley-type background and the calculation of the elemental composition was performed on the basis of Scofield’s relative sensitivity factors. This data system has an integrated library of peak positions and relative intensities. Identified peaks are automatically added to a peak table which can then be used for quantification. For quantification purposes, the data system has integral libraries of relative sensitivity factors (both Scofield and Wagner) and all data files contain information about the acquisition conditions, including the spectrometer conditions. The quantification routine automatically applies the correct instrument transmission function to the data.

  • Any visible light associated applications will enhance the quality of the further.

Response: Visible light associated applications of alumina-carbon films not studied yet. We address these investigations in future.

Minor comments:

  • Please take care of subscripts and superscripts throughout the manuscript (Ex: line 70, 76, 77, etc)

Response: The indicated mistakes corrected. Thank you for this remark.

  • All references must be in square brackets. check line 93.

Response: This mistake corrected to [23,24]

Response to Reviewer #2:

The bulk alumina always possesses a very high bandgap and shows no applications in visible light. A. Jagminas et al have successfully prepared alumina films with visible light absorption. Present work is interesting and very well supported by characterization methods. I recommend this work for publication.

Major comments:

  • How do authors calculate at% from XPS? It would be helpful for readers to see any mathematical expression associated with the calculation.

Response: For clarity we placed the following info in the Experimental part: The spectra calibration, processing and fitting routines were done using Avantage software (5.918) provided by Thermo VG Scientific. Core level peaks of Al2p, V2p, C1s and O1s were analyzed using a nonlinear Shirley-type background and the calculation of the elemental composition was performed on the basis of Scofield’s relative sensitivity factors. This data system has an integrated library of peak positions and relative intensities. Identified peaks are automatically added to a peak table which can then be used for quantification. For quantification purposes, the data system has integral libraries of relative sensitivity factors (both Scofield and Wagner) and all data files contain information about the acquisition conditions, including the spectrometer conditions. The quantification routine automatically applies the correct instrument transmission function to the data.

  • Any visible light associated applications will enhance the quality of the further.

Response: Visible light associated applications of alumina-carbon films not studied yet. We address these investigations in future.

Minor comments:

  • Please take care of subscripts and superscripts throughout the manuscript (Ex: line 70, 76, 77, etc)

Response: The indicated mistakes corrected. Thank you for this remark.

  • All references must be in square brackets. check line 93.

Response: This mistake corrected to [23,24]

Reviewer 3 Report

The manuscript is interesting and could be published after minor revision according to the comments below:

  1. Page 2, line 55 - The authors use the abbreviation PAA without any preliminary explanation.
  2. Page 3, line 94 - "cold solutions" - what is the temperature?
  3. Figure 2 - "0.25 NaVO3"? Same for page 4, line 138 and on other places in the text.
  4. Page 4, line 144 - "0.2 NaVO3 + 0.8 mol L-1 HCOOH"; Figure 8 - "0.2 M NaVO3 and 0.8 M HCOOH". The case need to be clarified.
  5. Figure 4 - "XP spectra" should be XPS spectra - appears also in other places.

Author Response

Response to Reviewer 3:

The manuscript is interesting and could be published after minor revision according to the comments below:

  1. Page 2, line 55 - The authors use the abbreviation PAA without any preliminary explanation.

Response: We add explanation.

  1. Page 3, line 94 - "cold solutions" - what is the temperature?

Response: All anodization experiments performed at ambient temperature of about 20°C as stated in the Experimental part. Therefore, we corrected this sentence by indication …’at room temperature’

Figure 2 - "0.25 NaVO3"? Same for page 4, line 138 and on other places in the text.

Response: The concentrations of solutions checked and corrected throughout the paper.

  1. Page 4, line 144 - "0.2 NaVO3 + 0.8 mol L-1 HCOOH"; Figure 8 - "0.2 M NaVO3 and 0.8 M HCOOH". The case need to be clarified.

Response: Thanks to this remark, we corrected the units.

  1. Figure 4 - "XP spectra" should be XPS spectra - appears also in other places.

Response: We changed XP spectra to XPS spectra, as required.

Reviewer 4 Report

In this manuscript,  Jagminas et al. reported a new way for the formation of 54
PAA film heterostructured with carbon-containing species at significantly lower anodizing voltage and current density with even a larger content of entrapped carbon species. This work is generally novel. But there are some experimental design and presentation issues that the authors should address.

  1. I suggest that the authors should provide more information in the figures. E.g. in Figure 2 and 6, note the different parameters for each curve and note the peaks in XPS curves. It will make the reader easier to understand the content. What's more, the resolution of some figures should be improved.
  2. I suspect that there is no control experiments in the design. There authors may want to consider using their previous strategy in ref [15] as the control or use the solution without NaVO3 as the control since it's crucial for the proposed strategy. Another thing is that the authors can also consider to provide a form summarizing the differences between the current anodized films with the control films on e.g. film morphology, film evenness, voltage, carbon species, etc. The authors can then get a more straightforward and comprehensive idea on how this work is pushing this field forward.
  3. The authors mentioned the film evenness several times in the manuscript. But there is no quantitative analysis. I suggest the authors performing AFM to get some data to support this point.
  4. The authors mentioned the existence of graphene. But there is some problems in confirming it. I am a little bit curious about it. I believe the graphene will also exist on the surface part. Thus, Raman spectroscopy will be able to characterize it. The authors may want to try it.
  5. How will this film lead to more practical applications? Can the authors enlighten the readers in the introduction part or conclusion part?

Author Response

Response to Reviewer #4:

To Editor of Materials                                                                           Vilnius, 23th March, 2022

                                                                                                                   Manuscript resubmission

                                                                                                                    ID: materials-1645635

Dear Editor,

     We are submitting the revised version of our paper titled “Designing carbon enriched alumina films possessing a visible light absorption”.  A specialist corrected English, Figs. 1a, 5, 6, and 8 were reconstructed according to requirements of Ref. #1, and new Table was inserted.    All changed places are signed by yellow. All authors have read it and agreed to its submission.

In this manuscript, Jagminas et al. reported a new way for the formation of 54
PAA film heterostructured with carbon-containing species at significantly lower anodizing voltage and current density with even a larger content of entrapped carbon species. This work is generally novel. But there are some experimental design and presentation issues that the authors should address.

  1. I suggest that the authors should provide more information in the figures, e.g. in Figure 2 and 6, note the different parameters for each curve and note the peaks in XPS curves. It will make the reader easier to understand the content. What's more, the resolution of some figures should be improved.

Response: The resolution of figure plots was improved. For clarity, the peaks of XPS curves presented in a new Table 3.

2.I suspect that there is no control experiments in the design. There authors may want to consider using their previous strategy in ref [15] as the control or use the solution without NaVO3 as the control since it's crucial for the proposed strategy. Another thing is that the authors can also consider to provide a form summarizing the differences between the current anodized films with the control films on e.g. film morphology, film evenness, voltage, carbon species, etc. The authors can then get a more straightforward and comprehensive idea on how this work is pushing this field forward.

Response: The peculiarities of Al anodizing in the aqueous solutions of formic acid without NaVO3 were investigated and results are presented in a separate paper. Currently, it is in the stage of minor revision of Trans Inst Metal Finishing journal.

3. The authors mentioned the film evenness several times in the manuscript. But there is no quantitative analysis. I suggest the authors performing AFM to get some data to support this point.

Response: The morphology of alumina films formed in the mixed solution of NaVO3 and HCOOH at the optimized composition is shortly discussed in the 3.4 section. Back-side SEM images shown in (a) and (b) panels of Figure 1 clearly show uneven growth of these films because the groups of several in times larger cells and pores are viewed at the film½metal interface. It is questionable if AFM images of interfaced surface at a sub-micrometer scale could be an additional verification of disorder growth. We will check this method in the next study. 

4. The authors mentioned the existence of graphene. But there is some problems in confirming it. I am a little bit curious about it. I believe the graphene will also exist on the surface part. Thus, Raman spectroscopy will be able to characterize it. The authors may want to try it.

Response: We agree that Raman spectra can be useful for determination of graphene in a pure state. But this is doubtful for mixed carbonaceous compounds in the alumina matrix. However, we thank reviewer for this suggestion and apply the Raman in the future study.    

5.How will this film lead to more practical applications? Can the authors enlighten the readers in the introduction part or conclusion part?

Response: In the Introduction part we mentioned that alumina thick films containing a high amount of entrapped carbonaceous species could be prospective for absorption of microwaves and fabrication of invisible objects. However, we address these questions for further investigations.

Round 2

Reviewer 4 Report

I suggest accepting the manuscript in the current form.

Author Response

Thank you very much for your review.